# Deep Learning for Super-Resolution of Mediterranean Sea Surface Temperature Fields

Claudia Fanelli[1*], Daniele Ciani[2], Andrea Pisano[2], and Bruno Buongiorno Nardelli[1]

[1]Consiglio Nazionale delle Ricerche, Istituto di Scienze Marine (CNR-ISMAR), Calata Porta di Massa, 80133, Naples, Italy
[2]Consiglio Nazionale delle Ricerche, Istituto di Scienze Marine (CNR-ISMAR), Via del Fosso del Cavaliere 100, 00133, Rome, Italy

**Correspondence:** Claudia Fanelli (claudia.fanelli@cnr.it)

**Abstract.**

Sea surface temperature (SST) is one of the essential variables of the Earth climate system. Being at the air-sea interface, SST modulates heat fluxes in and out of the ocean, provides insight into several upper/interior ocean dynamical processes, and it is a fundamental indicator of climate variability potentially impacting the health of marine ecosystems. Its accurate estimation and regular monitoring from space is therefore crucial. However, even if satellite infrared/microwave measurements provide much better coverage than what is achievable from in situ platforms, they cannot sense the sea surface under cloudy/rainy conditions. Large gaps are present even in merged multi-sensor satellite products and different statistical strategies have thus been proposed to obtain gap-free (L4) images, mostly based on Optimal Interpolation algorithms. These techniques, however, filter out the signals below the space-time decorrelation scales considered, significantly smoothing most of the small mesoscale and submesoscale features. Here, deep learning models, originally designed for single image Super Resolution (SR), are applied to enhance the effective resolution of SST products and the accuracy of SST gradients. SR schemes include a set of computer vision techniques leveraging Convolutional Neural Networks to retrieve high-resolution data from low-resolution images. A dilated convolutional multi-scale learning network, which includes an adaptive residual strategy and implements a channel attention mechanism, is used to reconstruct features in SST data at $1/100°$ spatial resolution starting from $1/16°$ data over the Mediterranean Sea. The application of this technique shows an improvement in the high resolution reconstruction, capturing small scale features and providing a root-mean-squared-difference improvement of $0.02°C$ with respect to the L3 ground-truth data.

## 1 Introduction

Investigating ocean dynamics and climate variability requires accurate, regular and systematic observations of the Sea Surface Temperature (SST). SST plays indeed a key role in air-sea interaction and upper ocean circulation processes (Warner et al., 1990; Deser et al., 2010; Chang and Cornillon, 2015), it is used to track climate variability and change (Jha et al., 2014; Pisano et al., 2020), and it is at the base of various chemical and biological processes (MacKenzie and Schiedek, 2007; Dong et al., 2022a). SST and the estimate of its gradients have also been proven to be a powerful tool to assess and investigate mesoscale and submesoscale variability (e.g., Bowen et al., 2002; Isern-Fontanet et al., 2006; González-Haro and Isern-Fontanet, 2014; Rio

et al., 2016; Castro et al., 2017; Ciani et al., 2020). Therefore, the availability of high resolution SST fields is crucial, since they serve as the primary data source for many scientific and operational applications. However, their reliability is hindered by the limitations of infrared (IR) and microwave-based (MW) measurements (Minnett et al., 2019). In fact, thermal IR instruments are able to provide SST images at kilometer to sub-kilometer scale resolution, although their application is limited by cloud cover, aerosols radiation emission/absorption and scattering. Conversely, SST retrieval in the microwave is hampered only due to sunglint, rain, radio frequency interference or proximity to land, but the lower spatial resolution achievable with present platforms represents a significant disadvantage (for instance, the footprint of each sample of the NASA Advanced Microwave Scanning Radiometer for EOS is an ellipse of approximately 45x64 kms). Higher resolution MW SSTs ($\simeq$ 15 km resolution) will only be available after the launch of the Copernicus Imaging Microwave Radiometer (Pearson et al., 2019), expected during 2029. Consequently, SST fields at high resolution are generally affected by several data voids. For this reason, a few statistical techniques have been developed to obtain gap-free SST images, mostly based on optimal interpolation (OI) (Bretherton et al., 1976). However, as a result of the temporal and spatial averaging applied during the interpolation, the effective resolution of the interpolated products can be significantly coarser than the nominal grid resolution, rarely getting down to less than a few tens of kilometres (Chin et al., 2017; Ciani et al., 2020; Yang et al., 2021). As such, providing interpolated data increases the accessibility of sea surface temperature fields for a wide community of users, but this improvement comes with a trade-off, as statistical interpolation leads to a strong smoothing of small scale ocean features.

In this context, we investigate here the potential of applying deep learning models to improve the effective resolution of a gap-free SST field, providing those small scale features even when direct measurements are missing. We exploit techniques generally used in the field of computer vision, which have proven to be very successful especially for processing gridded data, managing large-scale datasets while controlling the computational efficiency. In the field of oceanography, the research community started only recently to explore the applicability of machine learning (ML) methods to ocean remote sensing images (Dong et al., 2022b). The applications range from oil spill (Singha et al., 2013) to eddy detection (Lguensat et al., 2018; Duo et al., 2019) and parametrization (Bolton and Zanna, 2019), to marine algae species discrimination (Balado et al., 2021; Cui et al., 2022), to forecasting of ocean variables (Deo and Naidu, 1998; Ham et al., 2019) and estimation of meteorological parameters (Krasnopolsky et al., 2013; Zanna et al., 2019). Moreover, good results have been achieved from applying Neural Networks (NNs) to space-time interpolation and short-term forecasting issues with satellite altimetry data (Fablet et al., 2021), to high-performance description of turbulence processes (Mohan et al., 2020; Zanna and Bolton, 2020) and to SST reconstruction (Meng et al., 2021; Lloyd et al., 2021).

Among image processing techniques, impressive results have been obtained with Convolutional Neural Networks (CNNs) due to their high ability to extract the most important information from two-dimensional spatial fields. Recently, the application of CNN architectures in the process of reconstructing high-resolution images from low-resolution ones, the so-called single image Super Resolution (SR) problem, has attracted much attention in a wide range of scientific challenges. The idea is to implement a network that directly learns the end-to-end mapping between low and high resolution images. One of the simplest attempts made by Dong et al. (2015) was the construction of a network for image restoration composed of three 2D convolutional layers with different kernel sizes. The three layers might be seen as the three conceptual phases of this

NN algorithm: a first extraction of overlapping tiles from the input images representing them into feature maps; the non-linear mapping of these maps onto one high-dimensional vectors of high-resolution feature maps; the final reconstruction aggregating the above representations to generate the final high-resolution image. This final image is expected to be similar to the ground truth one. Despite the simplicity of the architecture, the SRCNN developed by Dong et al. (2015) achieved excellent results with respect to more traditional methods and it has already been applied to reconstruct satellite-derived SST data, with promising results (Ducournau and Fablet, 2016). Building on Dong's work, several more complicated structures have been developed to tackle the Super Resolution problem. In subsequent years, a few attempts to develop residual networks have shown the convergence improvement of deeper architectures, mainly given by the introduction of skip-connections and recursive convolutions (He et al., 2016; Kim et al., 2016a, b). Similarly, a step forward has been made by Lim et al. (2017) with the development of an Enhanced Deep Residual Network for Super Resolution (EDSR) which makes use of residual blocks with constant scaling layers after the last convolutional layer, in order to stabilize the training even in presence of a large number of filters. This modification led to significantly better accuracy using much deeper networks, while controlling the computational cost of the training phase. A further step has been proposed by Liu et al. (2019) with the Adaptive Residual Blocks (ARBs), which replace the constant factor with adaptive residual factors, increasing the adaptability of the network. More specifically, in the ARB, feature responses (i.e., filter output channels) are re-calibrated on a channel-wise basis using a so-called Squeeze and Excitation module, before being combined with the block input. This process is in fact able to enhance the network's capability to capture intricate relationships among the learned feature channels. Recently, to further push these networks to efficiently handle different spatial scales in a multichannel input (each channel including a different variable with characteristic feature scales), Buongiorno Nardelli et al. (2022) introduced dilated convolutional multi-scale learning modules in the network developed by Liu et al. (2019), expanding the network receptive fields while still controlling its computational cost. In that work, the deep learning network was designed to super-resolve absolute dynamic topography (ADT) learning from both low resolution ADT and high resolution SST data, through an observing system simulation experiment (namely simulating all observations through an ocean general circulation numerical model).

Our aim is to exploit the ability of the dilated SR network to increase gap-free SST effective resolution, directly training our network on remote sensing SST data for both the input and target datasets. We make use of the data produced by the Italian National Research Council - Institute of Marine Sciences (CNR-ISMAR), within the Copernicus Marine Service, consisting of merged multi-sensor (L3S) and gap-free (L4) Sea Surface Temperature products over the Mediterranean Sea at high resolution (HR, nominally $1/16°$) and ultra-high resolution (UHR, nominally $1/100°$) (Buongiorno Nardelli et al., 2013). Considering that our UHR interpolation accounts for space-time decorrelation scales of 5 km and 1 day, in the presence of valid L3 UHR observations, the resulting L4 data can be considered as submesoscale resolving (Kurkin et al., 2020). UHR OI processing makes use of upsized L4 HR fields as an initial guess. This guess is left unchanged in the absence of L3 UHR data and the L4 effective resolution is thus by definition lower than $1/100°$ there. Therefore, our goal is to enhance the effective resolution of the gap-free upsized $1/16°$ background field.

## 2 Materials and Methods

### 2.1 Training and test datasets

When dealing with deep learning methods it is important to construct the training and test datasets, to ensure a sufficient generalization capability and, more specifically, to avoid under and over-fitting problems. Under-fitting occurs when the model fails to achieve a suitably low error on the training set, while over-fitting occurs when the gap between the training error and test error becomes excessively wide (Goodfellow et al., 2016). Remote sensing data are a very suitable resource to prevent the occurrence of these problems, due to the wide availability of large-scale gridded datasets which are complex enough to encapsulate an extensive variety of features.

The suite of products considered for this project provides the foundation SST (i.e., the temperature free of diurnal variability) over the Mediterranean Sea from 2008 to present (https://doi.org/10.48670/moi-00172) at Near Real Time (NRT). These data are built from level 2 (L2) infrared measurements (i.e., data in satellite native grid/swath coordinates) processed through a series of steps divided into different modules as detailed by Buongiorno Nardelli et al. (2013) and sketched in Figure 1.

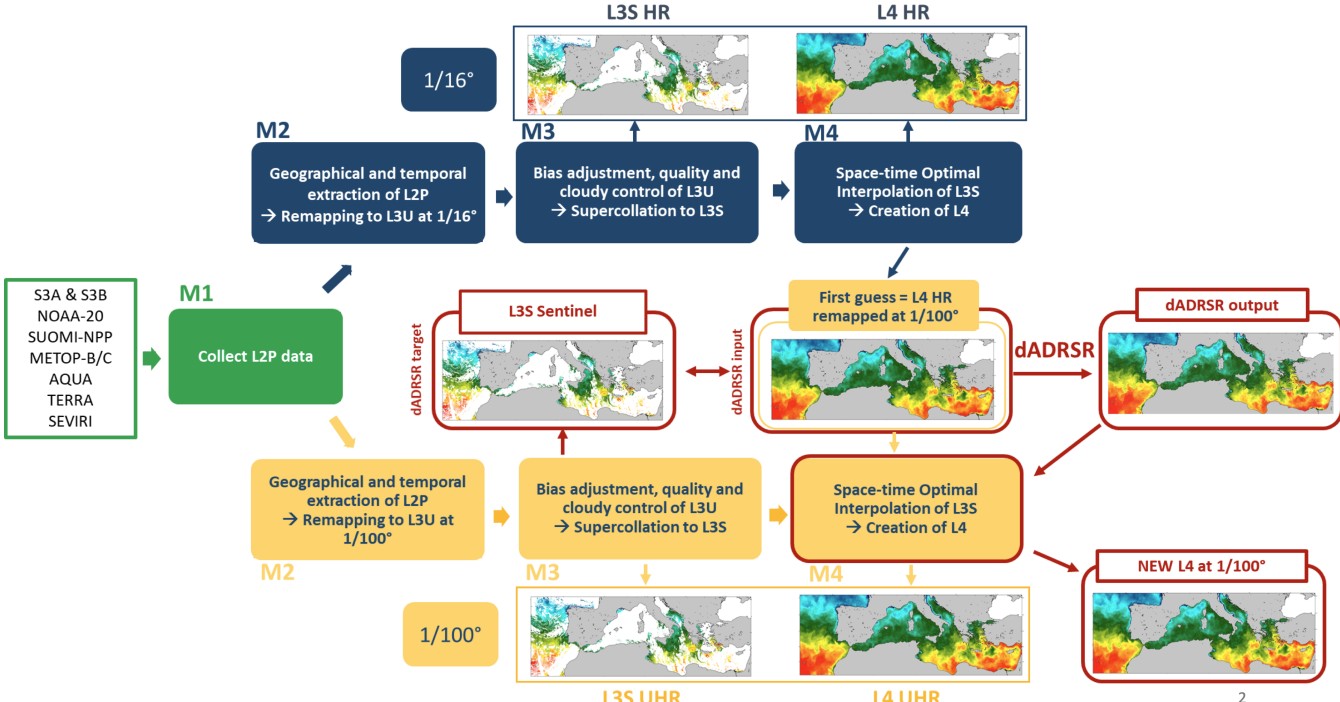

**Figure 1.** Sketch of the processing chains for the production of L3S and L4 SST fields at 1/16° and 1/100° spatial resolution over the Mediterranean Sea. Green boxes represent shared modules, blue boxes contain the information and the data for the HR product, yellow boxes refer to the UHR processing chain and red boxes highlight the application of the super-resolution technique.

The L2 measurements are obtained from several instruments on board both geostationary and polar orbiting satellites (including Sentinel-3A and Sentinel-3B, NOAA-20, SUOMI NPP, Metop-B, Metop-C, AQUA, TERRA and SEVIRI). After a first shared module for the upstream data collection (M1), two separated processing chains are used to obtain the SST products at $1/16°$ and $1/100°$ spatial resolution. Both include a second module for the geographical and temporal extraction of the L2 data and the remapping onto the corresponding regular grid (M2), and a third module for the bias correction and quality control which lead to the merging of all the data from the different sensors (M3). The output of each of these processing paths is an L3S product at the associated spatial resolution. The final gap-free fields are provided by applying a space-time Optimal Interpolation algorithm to the L3 data for both HR and UHR products (M4). However, the two OI schemes use different initial guesses in absence of the observed data. While the HR processing chain makes use of a climatological background field to produce the L4 SST image at $1/16°$ resolution, in the case of the finer resolution product the initial guess used by the OI algorithm is the L4 HR SST field, preliminary upsized onto a $1/100°$ regular grid (through a thin plate spline). As a consequence, due to the small decorrelation scales assumed in the UHR interpolation, small scale features are correctly represented only when valid UHR L3 observations are present close to the interpolation point within a short temporal window. For this reason, in the absence of these observations, the final UHR product will have an effective resolution equal to or coarser than $1/16°$.

We train the network to improve the satellite SST effective resolution introducing realistic small scale features in the interpolated and upsized gap-free L4 HR SST images (dADRSR input in Fig. 1). The target data are derived from a ground-truth super-collated L3S UHR SST dataset (namely merged multi-sensor data) specifically built for this purpose. The dataset (called L3S Sentinel, hereinafter) is obtained by applying the first three modules of the UHR SST processing chain to acquisitions only from the Sea and Land Surface Temperature Radiometer (SLSTR) on board of the Sentinel 3A and 3B satellites, due to their high radiometric accuracy and km-scale resolving capabilities (Coppo et al., 2020).

Both input and target datasets are mapped on a regular grid at $1/100°$ spatial resolution over the Mediterranean Sea for the year 2020. Since our goal is to retrieve small scale features, a first moving average high pass filter (with a kernel radius of 200 km) is applied to remove the large scale dynamics in both input and target images (Kurkin et al., 2020). Then, all 100x100 km regions (referred to as tiles herein) consisting of at least 95% valid values were extracted from the filtered fields. Tiles were allowed to overlap at most one other tile by 50%. SST values are then transformed into anomalies to avoid seasonal variability and scaled between -1 and 1 through a classical min-max normalization technique, i.e.:

$$SST_{norm} = 2\left(\frac{SST - \min(SST)}{\max(SST) - \min(SST)}\right) - 1. \tag{1}$$

The test dataset is finally selected setting aside four individual days which are representative of the different seasons and composing the 15% of the total number of the tiles available after the preprocessing. These data are separated in a fully independent dataset (i.e., data never used in the training phase) which is composed by 18576 overlapped tiles. The training dataset finally consists of 94110 pairs of tiles extracted from the rest of the 362 days, of which one part (properly separated from the original training dataset at the beginning of the training phase, following a 85:15 ratio between the two) is used by the network in the validation phase. In Figure 2 an example of the images used to construct the two datasets is shown. The top left panel depicts the gap-free first guess map (i.e., L4 HR upsized at $1/100°$ spatial resolution) on 4th January 2020 and below the

corresponding L3S Sentinel image derived from merged Sentinel 3A and 3B data. On the right, we provide an example of the

140 a pair of tiles extracted from the related low and high resolution SST, fed as input and target to train the network, respectively.

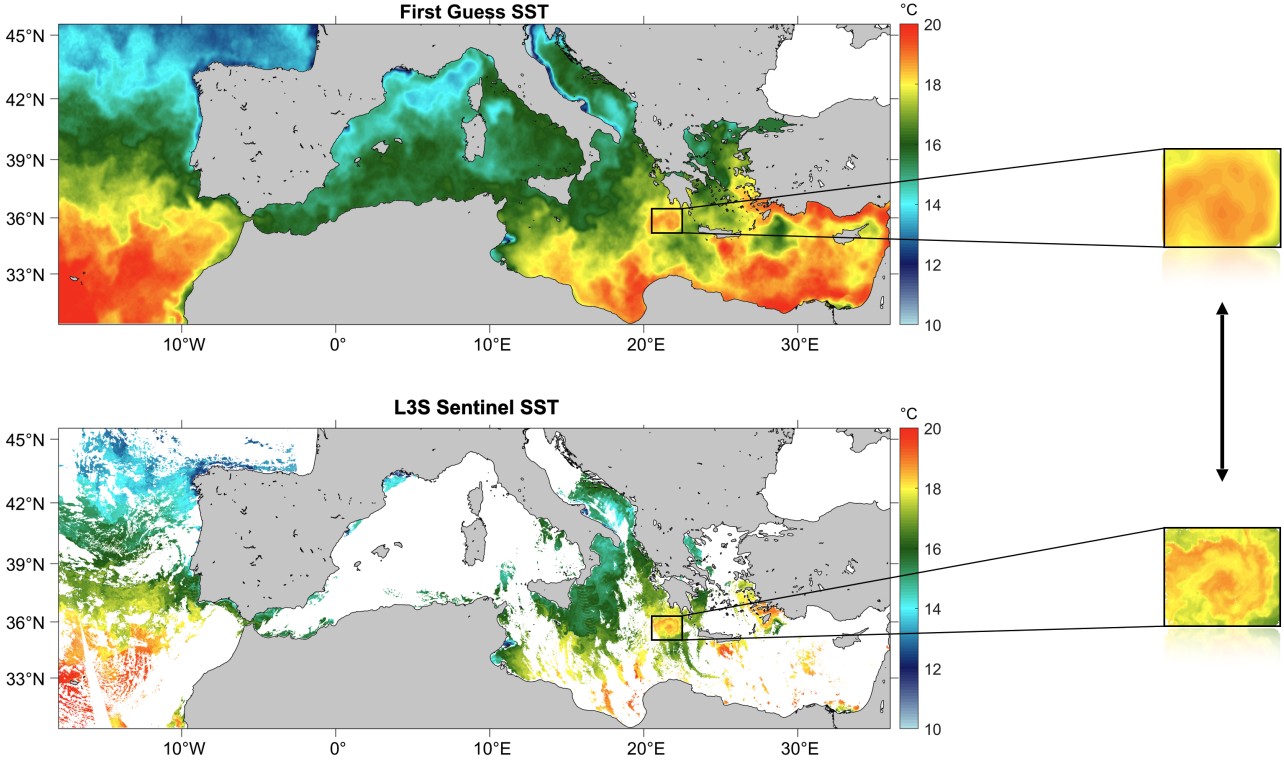

**Figure 2.** On the left the SST First Guess map used to extract the input tiles (top) and the SST L3S Sentinel target image (bottom) on 4th January 2020. On the right an example of an extracted tile used for the training and test datasets.

## 2.2 Super Resolution Convolutional Neural Network

In deep learning, Super Resolution algorithms are example-based methods, which generate exemplar patches from the input image. As mentioned above, the application of Convolutional Neural Networks to the Super Resolution problem is based on networks that directly learn an end-to-end mapping between low and high resolution images. These networks consist of a series

of interconnected layers, which make use of the convolution operator simulating the connectivity between neurons observed in the organization of the animal visual cortex. Formally, the output $Y$ of each layer $i$ is a function of a transformation of the previous layer output $X$, i.e.,

$$Y = F(W_i * X + B_i), \tag{2}$$

where $F$ is the non-linear activation function (in this case the Rectified Linear Unit or ReLU), $W_i$ the weights, $B_i$ the biases and $*$ the convolution operator. The array of weights, generally called filters or kernels, is able to detect a specific type of feature in the input (generating what is called a features map) and might include for instance Gaussian-like filters or edge detectors along different directions, or any other kind of filter learned during the training. Having its own functionality, each layer will contain different structures. The main idea is that the network learns the weights of the filters and updates the parameters of the system through an optimization process based on minimizing the error between the output and the data from the validation set. This simple yet powerful idea can be augmented as much as desired, especially exploiting the potential of deep networks. The CNN implemented here was originally developed by Buongiorno Nardelli et al. (2022) and called dilated Adaptive Deep Residual Network for Super-Resolution (dADRSR). In the dADRSR network (schematized in Fig. 3) the low resolution input dataset is initially fed to three parallel dilated convolutional layers with the same number of filters (equal to 3 $\times$ 3) but increasing dilation factor (1, 3 and 5, respectively), which allows for a larger receptive field, extracting information at different scales without increasing the number of parameters. After this first stage, the data pass through a sequence of twelve Multiscale Adaptive Residual Blocks (M-ARB), each including two sets of parallel dilated convolutional layers (with 120 and 10 filters, respectively) and a Squeeze-and-Excitation (SE) module able to improve channel interdependencies at almost no computational cost (Hu et al., 2018), before being summed up to produce the final high resolution output. The main characteristics of an adaptive strategy involves replacing the fixed scaling of learned features with an adaptive scaling mechanism, carried out by the SE block (which is a kind of attention mechanism). It captures the global importance of each channel by initially squeezing the feature maps to a single numeric value (therefore obtaining a vector of size equal to the number of channels) and, finally, feeding this output to a two-layer "bottleneck" network which will produce new features maps, scaling each channel based on its importance. The detailed discussion of the architecture of the network may be found in Buongiorno Nardelli et al. (2022). The training algorithm follows an early stopping rule which terminates the iterations as soon as the validation loss function increases for a previously chosen number of epochs (defined by the *patience* parameter which is set to 20 here). An adaptive learning rate (initialized at $l_r = 10^{-4}$) is given by the Adam optimizer (Kingma and Ba, 2014), where the hyperparameters are set following the values found in most of the recent literature (Lim et al., 2017; Liu et al., 2019; Buongiorno Nardelli et al., 2022): the numerical stability constant is $\varepsilon = 10^{-8}$, the exponential decay rates for the first and the second moment estimates are set to $\beta_1 = 0.9$ and $\beta_2 = 0.999$, respectively. Instead of using the classical dropout regularization technique, a DropBlock strategy is implemented where contiguous regions of a feature map are dropped together, which has been shown to increase the accuracy of the network for convolutional layers (Ghiasi et al., 2018). To evaluate the accuracy, the mean-squared error is used as reference in the loss function. The dADRSR training model finally uses almost 1.6 M trainable parameters. All codes are written in Python using the deep learning framework *Keras* and the training was performed on a single NVIDIA T4 GPU in almost 4 days.

## 2.3 Evaluation of model performances

Three different error measures are calculated to evaluate the network reconstruction performance between the ground-truth image $x$ and the network output $y$.

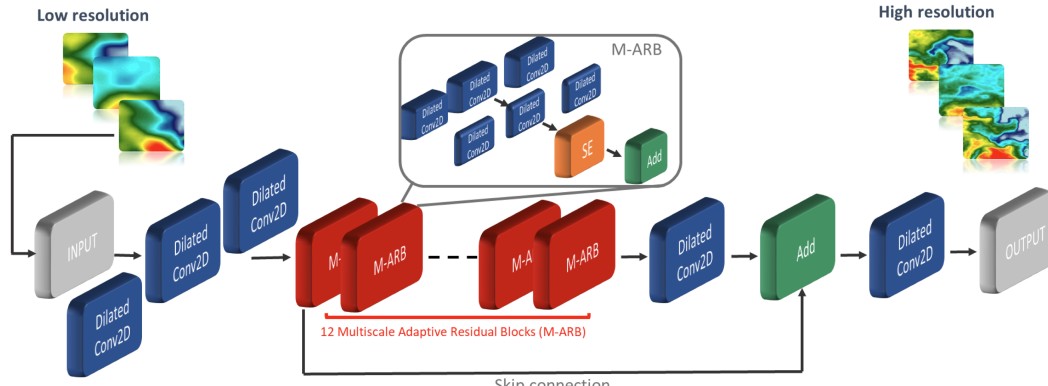

**Figure 3.** Schematic of the dilated Adaptive Deep Residual Network for Super-Resolution developed by Buongiorno Nardelli et al. (2022). Conv stands for convolutional; M-ARB for Multiscale Adaptive Residual Block; SE for Squeeze and Excitation module; and Add indicates aggregations of the outputs from the networks blocks.

Firstly, we consider the classical Root Mean Squared Error (RMSE) given by:

$$RMSE(x,y) = \sqrt{\frac{\sum_i^N (x_i - y_i)}{N}}, \tag{3}$$

where $x_i$ and $y_i$ are the SST values of the $i$th pixel of the images $x$ and $y$, respectively, and $N$ is the total number of pixels. This measure can be useful to evaluate the accuracy of a reconstructed value pixel by pixel, but it can be misleading in the assessment of the ocean state reconstruction, depending on the objectives of the application considered. For instance, if the NN is able to reproduce an ocean structure in a position which is slightly misplaced with respect to the ground-truth measurement, the RMSE will be high indicating a poor reconstruction, even worse than if it would have entirely missed the structure. This issue is often referred to as the double penalty issue, since point-matching measures will penalize the misplacement twice (where the structure should actually be and where it is incorrectly predicted). However, in some cases it is possible that capturing an ocean phenomenon, even if in a slightly wrong position, is better than missing it completely. For this reason, we also consider two additional measures usually considered in image processing.

The most commonly used measure for reconstructed image quality is the Peak Signal to Noise Ratio (PSNR), representing the ratio between the maximum possible pixel value of the image $I$ and the power of distorting noise that affects the quality of its representation, usually represented by the RMSE itself:

$$PSNR(x,y) = 20 \log_{10}\left(\frac{\max(I)}{RMSE(x,y)}\right). \tag{4}$$

The PSNR can be seen as an approximation to human perception of reconstruction quality, where the higher the value the better the quality of the image.

The third error measure is the structural similarity index measure (SSIM) proposed by Wang et al. (2004), widely used for measuring image quality and especially the similarity between two images. The concept is based on the idea that while PSNR estimates perceived errors to quantify image degradation, SSIM captures changes in perceived structural information variation. That is, if a reconstructed image is altered with a different type of degradation (for instance, mean-shifted, blurred or with a salt-pepper impulsive noise effect), while the MSE will come out nearly identical for all the cases, the SSIM will capture the different perceptual quality, being a weighted combination of luminance, contrast and structure measurements. The mathematical formulation of the SSIM between two images $x$ and $y$ is given by:

$$SSIM(x,y) = \frac{(2\mu_x\mu_y + c_1)(2\sigma_{xy} + c_2)}{\left(\mu_x^2 + \mu_y^2 + c_1\right)\left(\sigma_x^2 + \sigma_y^2 + c_2\right)}, \tag{5}$$

where $\mu_x$ and $\mu_y$ are the mean values of $x$ and $y$, respectively, $\sigma_{xy}$ is the cross-correlation of $x$ and $y$, $\sigma_x^2$ and $\sigma_y^2$ are the variance of $x$ and $y$, respectively, and $c_1$ and $c_2$ are the regularization constants for the luminance, contrast, and structural terms.

All these errors are computed for the final reconstructed maps over the whole Mediterranean Sea. The combination of the super-resolved tiles is made considering a linear combination of the values obtained for the same pixel, weighted according to their position within the tile. That is, the coefficient of each value decreases as its distance becomes larger from the central pixel of the tile. Finally, the large scale field initially removed to isolate the small scale features is added back to reproduce the large scale structure of the original SST field.

## 3   Results and discussion

Our aim is to verify whether the dADRSR network, trained by means of satellite-derived observations, is able to improve the effective resolution of the SST fields in the areas where our interpolation technique removes most of the spatial variability associated with mesoscale and submesoscale processes. Due to computational costs, a preliminary study based on a restricted test dataset was performed. The performances of the dADRSR network in comparison with other deep learning algorithms and a sensitivity analysis on the impact of different choices for the architecture is presented in Sec. 3.1. Finally, an additional test on an independent dataset built on a one year long series of SST fields was carried out (Sec. 3.2).

### 3.1   Preliminary validation study

Figure 4 shows the comparison of the result obtained on one of the SST fields included in the test dataset which corresponds to the SST daily map of 1st August 2020. The reconstruction of the dADRSR network at 1/100° spatial resolution over the Mediterranean Sea is shown in the top panel. In order to visually evaluate the reconstruction of the network, the corresponding L3S SST merged field observed by Sentinel 3A and 3B (central panel) and the First Guess map used in the optimal interpolation algorithm (bottom panel) are shown below the network output. Evident from this figure is that the SST features estimated by the CNN appear much sharper than the ones approximated by the low resolution map, showing promising capabilities to effectively reconstruct dynamical features.

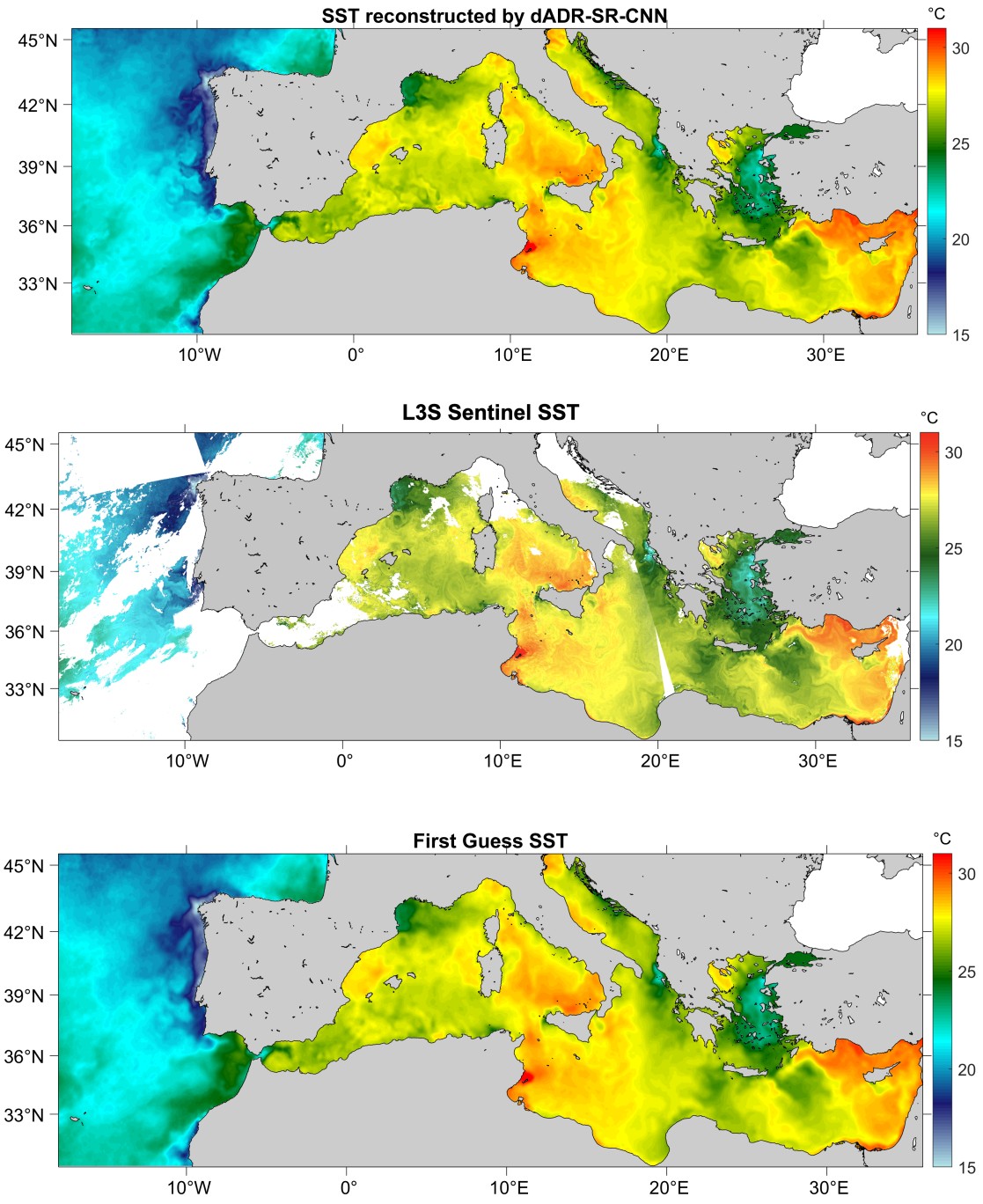

**Figure 4.** Comparison of the SST fields of 1st August 2020 provided by the reconstruction of the dADRSR (top panel), the L3S data measured by Sentinel 3A and 3B (central panel) and the First Guess SST field (bottom panel).

To highlight the ability of the CNN to more accurately capture small scale features with respect to the statistical algorithm, we show in Figure 5 three smaller panels corresponding to zoomed-in regions of the fields shown in Fig. 4 delimited by the coordinates [30, 40]°N and [19, 36]°E, with the correspondent SST spatial gradients (calculated using the Sobel operator) for the dADRSR reconstruction, L3S Sentinel ground-truth data and the first guess approximation (from top to bottom, respectively). This particularly structure-rich area is an excellent example to demonstrate the ability of the network to capture dynamical

processes which are quite clear in the high resolution L3S Sentinel data (shown in the central panels). In fact, SST fronts are strongly connected with the surface dynamics and they are generally associated with energetic motions at the mesoscale and submesoscale. While the CNN is able to capture at least the most energetic structures (as shown in the top panels of Fig. 5), the first guess presents an extremely smooth field due to the OI algorithm applied, where all the small scale features have been filtered out even when high resolution data are present. We recall that the Sobel gradient operates on a 3×3 pixel kernel, a

spatial scale not readily visible in the figures. This is why the first guess gradient field is so much weaker than the CNN or SLSTR fields even though the large scale structures are similar.

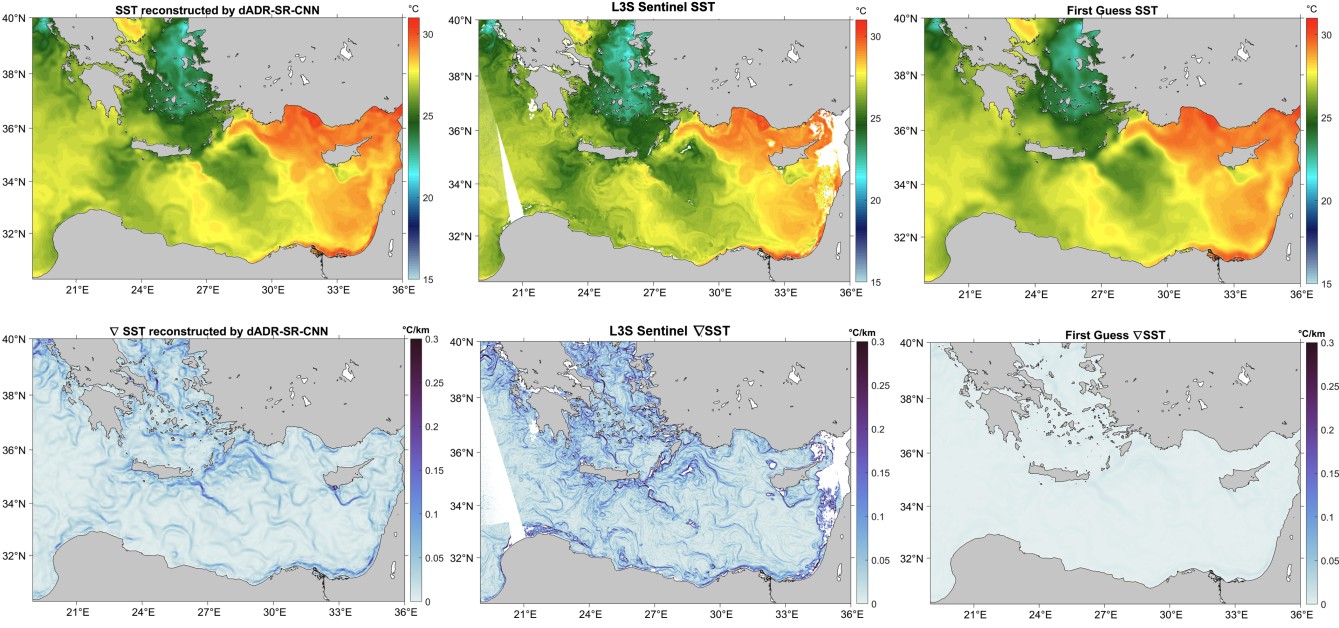

**Figure 5.** Comparison of the SST (on the left) and SST gradients (on the right) provided by the reconstruction of the dADRSR (top panels), the L3S data measured by Sentinel 3A and 3B (central panels) and the Optimal Interpolated First Guess (bottom panels) in the selected region delimited by the coordinates [30, 40]°N in latitude and [19, 36]°E in longitude.

This visual analysis is quantitatively confirmed by the maps in Figure 6, displaying the difference between the error made by the low resolution approximation and the dADRSR model with respect to the original L3S Sentinel image, averaged on 1°

× 1° boxes. Here, red indicates an improvement of the CNN with respect to the first guess image and blue a degradation. A clear predominance of red boxes is found both in the SST and the SST gradients error maps.

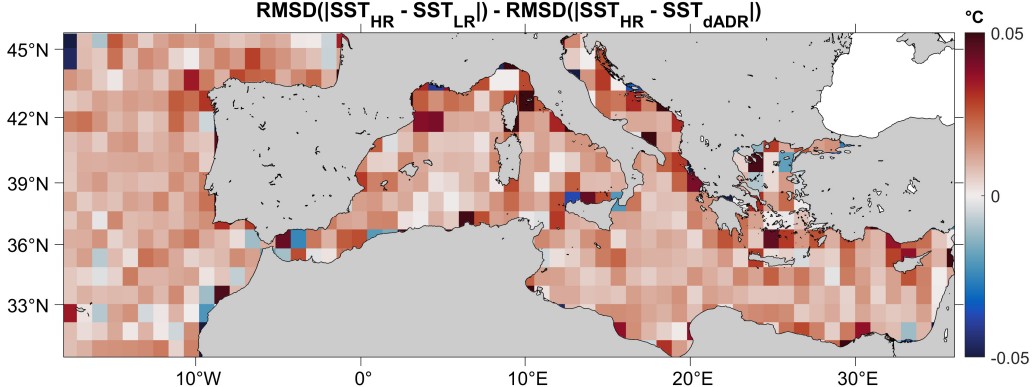

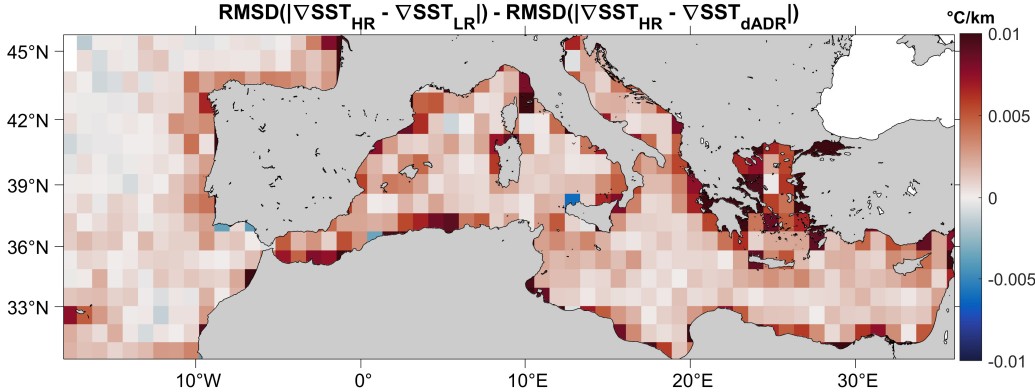

**Figure 6.** Comparison of the performance of the SST (top) and $\nabla SST$ (bottom) dADRSR reconstruction and the First Guess with respect to the L3S data measured by Sentinel 3A and 3B satellites. Red positive values show an improvement of the network reconstruction with respect to the optimal interpolated First Guess.

In Table 1, the comparison is summarized quantitatively, with the network reconstruction presenting a RMSE = 0.31°C (versus the 0.33°C obtained from the first guess approximation), a mean PSNR equal to 37.9 and a SSIM equal to 0.54, both

**Table 1.** Error estimations of the SST given by the First Guess map, the EDSR network (Lim et al., 2017), the ADR reconstruction (Liu et al., 2019), the dADRSR built with half of the M-ARB (called dADRSR/2) and the dADRSR output with respect to the L3S ground-truth: the RMSE given by Equations (3) and the corresponding confidence interval calculated by the bootstrapping procedure, the Peak-Signal to Noise Ratio obtained by (4) and the Structural Similarity Index Measure given by (5).

| Model | RMSE ($^\circ C$) | PSNR | SSIM |
|---|---|---|---|
| First Guess | $0.33 \pm 7 \times 10^{-5}$ | 37.5 | 0.53 |
| EDSR | $0.32 \pm 6 \times 10^{-5}$ | 37.7 | 0.54 |
| ADR | $0.32 \pm 6 \times 10^{-5}$ | 37.9 | 0.54 |
| dADRSR/2 | $0.32 \pm 8 \times 10^{-5}$ | 37.7 | 0.54 |
| dADRSR | $0.31 \pm 7 \times 10^{-5}$ | 37.9 | 0.54 |

larger than the first guess result. To highlight the quality of the dADRSR reconstruction we performed the same test using other deep learning super-resolution models: the Enhanced Deep Residual network for Super-Resolution (EDSR) proposed by (Lim et al., 2017), the Adaptive Deep Residual Network for Super-Resolution (ADR) developed by (Liu et al., 2019) and the dADRSR proposed by (Buongiorno Nardelli et al., 2022) setting the number of M-ARB equal to six instead of twelve (called dADRSR/2 hereinafter). We recall that while the RMSE should be low to ensure a good approximation, for the other two quantities high values indicate an improvement. The dADRSR output achieves the best value for all the evaluation methods presented.

To analyse the effectiveness of the high resolution reconstruction, we compare the Power Spectral Density (PSD) of the network output with the L3S Sentinel product and the First Guess map over three selected zones. We compute the PSD via Fast Fourier Transform (FFT) with a Blackman–Harris window over the three areas corresponding to the boxes with labels *a*, *b* and *c* in Figure 7. The zones are chosen in order to have the maximum number of valid pixels available in the L3S Sentinel observations and represent different dynamical regimes:

a - A region over the Sea of Sardinia of low spatial variability.

b - A region over the Ionian Sea with an important SST variability between the eastern and the western part of the area.

c - A region over the Levantine Sea characterized by small scale structures.

The three central panels of Figure 7 show the PSD, presented as a function of the wavenumber, of the SST reconstructed by the dADRSR (in yellow), the first guess map (in red) and the high resolution L3S Sentinel observations (in blue) over the three zones delimited by the black rectangles over the L3S SST field on the top panel. In all the cases note that the PSD follows the same behavior for wavenumbers smaller than 1 deg$^{-1}$, i.e., for scales larger than 100 km. This means that for such scales the SST fields obtained by both the CNN and the first guess characterize well large mesoscale features. Conversely, for regions *a* and *b*, both the first guess and the network reconstruction exhibit a significant PSD decrease for wavenumbers higher than 1 deg$^{-1}$, but starting from approximately 10 deg$^{-1}$, the first guess spectrum separates from the dADRSR spectrum (and

270 consequently increases the distance from the L3S Sentinel one), indicating a poorer reconstruction of spatial features below 10 km. The network spectrum, on the other hand, shows that the machine learning algorithm better captures those small scale features in all the cases. An analogous behaviour is found for the spectra of the SST gradients over the same three regions (three bottom panels). The abrupt decreases seen in the L3S Sentinel and first guess spectra starting at $3 \times 10 \ \text{deg}^{-1}$ are probably due to artifacts introduced by the re-gridding, as already discussed by Liberti et al. (2023), leading to the lack of physical meaning

for the spectra from this point onwards (i.e., for the highest wavenumbers). Overall, we can conclude that the CNN spectra tracks the observed spectra from $1 \ \text{deg}^{-1}$ to $10 \ \text{deg}^{-1}$ quite well while the first guess spectra are more energetic in this region. On the other hand, although the CNN spectra are less energetic than the L3S Sentinel spectra for wave numbers greater than $10 \ \text{deg}^{-1}$, they are more representative of the true spectra in this region than the first guess spectra.

To quantify the differences between the effective resolution of the products, we calculated the ratio between the spectral

content of the mapping error (e.g., $SST_{L3S} - SST_{dSDRSR}$) and the spectral content of the L3S Sentinel observation of the SST for the dADRSR reconstructed fields and the first guess map (see Figure 8) as follows:

$$RPSD(\lambda) = \frac{PSD_{\text{diff}}(\lambda)}{PSD_{\text{L3S}}(\lambda)}, \tag{6}$$

where $\lambda$ is the wavelength, $PSD_{\text{L3S}}$ is the spectra of the L3S Sentinel observations and $PSD_{\text{diff}}$ is the spectra of the differences between the L3S data and either the output of the CNN or the first guess field. The RPSD defined in Eq. (6) is

285 calculated over the three zones delimited by black rectangles on the L3S SST field on 1st August 2020 presented in the bottom panel of Figure 7. The effective resolution of the products, based on the intersection between this PSD ratio and the ratio equal to 0.5 as defined by (Ballarotta et al., 2019), shows the ability of the dADRSR network to resolve smaller scales.

In order to demonstrate the effective spatial resolution enhancement of the super-resolved images under cloudy conditions, we performed a power spectral density analysis on different SST products available over the Mediterranean Sea (Figure 9).

The products used are the L4 NRT HR at nominal $1/16°$ and the L4 NRT UHR at nominal $1/100°$ spatial resolution provided by CNR for the Copernicus Marine Service (https://doi.org/10.48670/moi-00172), the GLOBAL OCEAN OSTIA product developed by the UK MET OFFICE at $0.05°$ (https://doi.org/10.48670/moi-00165), the Multi-scale Ultra-high Resolution (MUR) product provided by the NASA-JPL at $0.01°$ (Chin et al., 2017) and the super-resolved SST field obtained by the application of the dADRSR network developed in this work at $0.01°$. The PSD are computed along the three transects (a, b

and c in the bottom panel of Figure 9), chosen in three areas affected (at different levels) by cloud coverage. In all the cases the slope of the spectra is very similar for the three products at higher resolution (namely the UHR, the MUR and the dADRSR approximations), while lower values are observed for the HR and OSTIA products (which do not reach wavenumbers larger than $0.1 \ \text{deg}^{-1}$ by construction). The PSD analysis for section (a) shows that the dADRSR approximation is not affected by excessive noise as are the other products, presenting PDS values that are higher than corresponding values of the MUR and

the UHR spectra (especially at small scales). For sections (b) and (c), the green line representing the dADRSR spectrum lies above all the other lines for almost the entire wavenumber range.

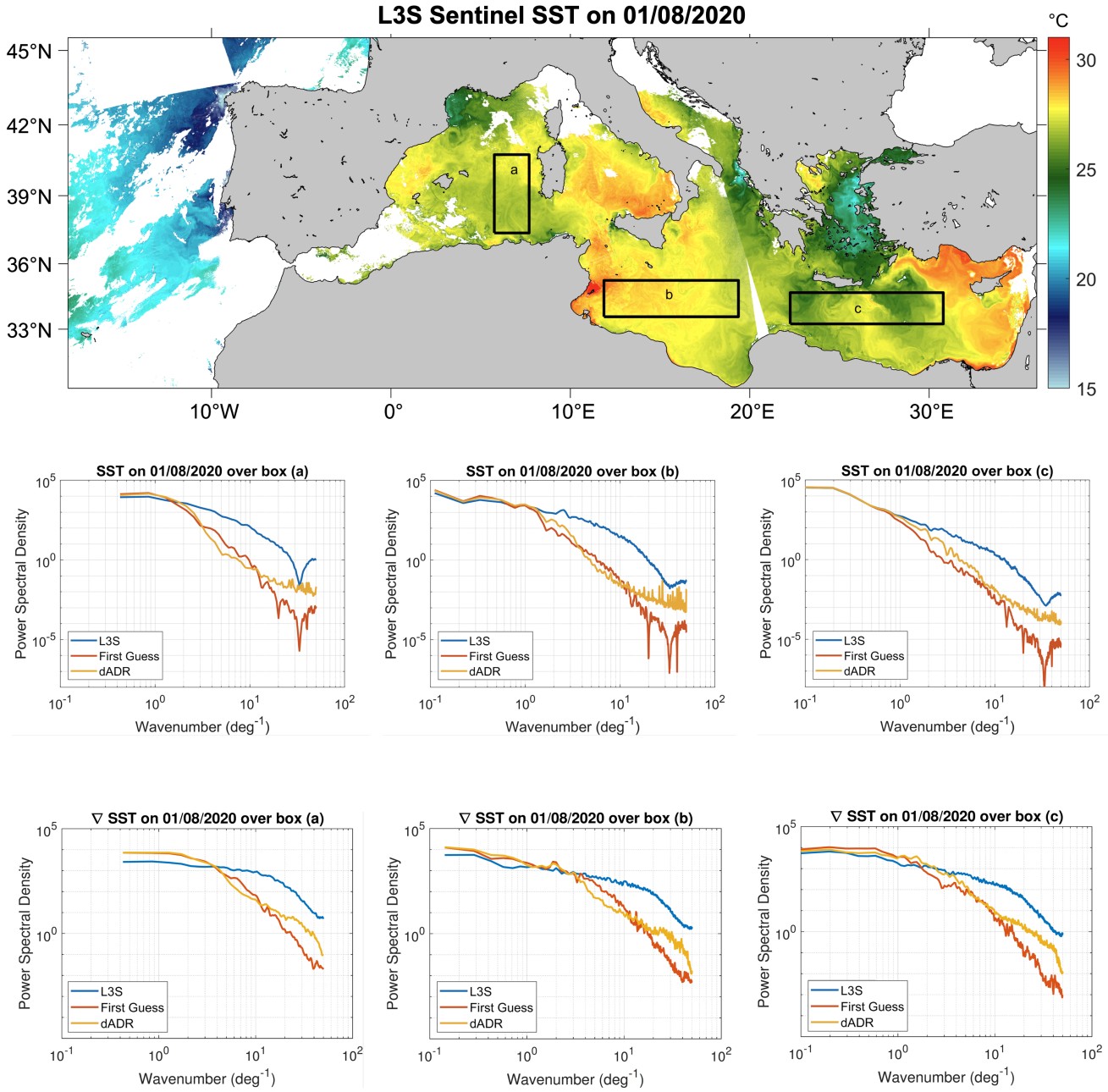

**Figure 7.** The PSD of the SST (central panels) and SST gradients (bottom panels) reconstructed by the dADRSR (in yellow), the first guess map (in red) and the L3S Sentinel high resolution observations (in blue) over the three zones delimited by black rectangles on the L3S SST field on 1st August 2020 in the top panel.

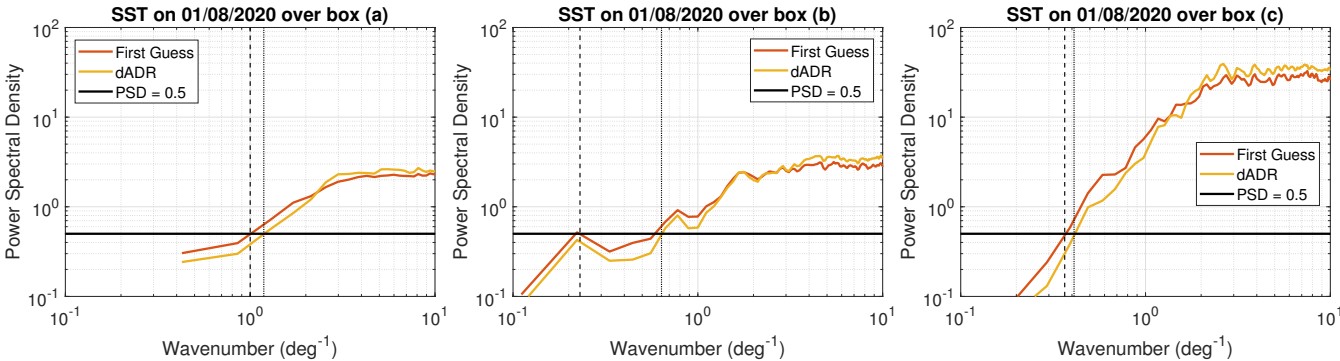

**Figure 8.** The ratio between the spectral content of the mapping error and the spectral content of the L3S observation of the SST for the dADRSR reconstructed fields (in yellow) and the first guess map (in red) over the three zones delimited by black rectangles on the L3S SST field on 1st August 2020 presented in the bottom panel of Figure 7. The black line represents the ratio equal to 0.5, the dashed black line and the dotted black line represent the intersection between PSD ratio of the first guess and the dADRSR, respectively, showing the effective resolution of the products.

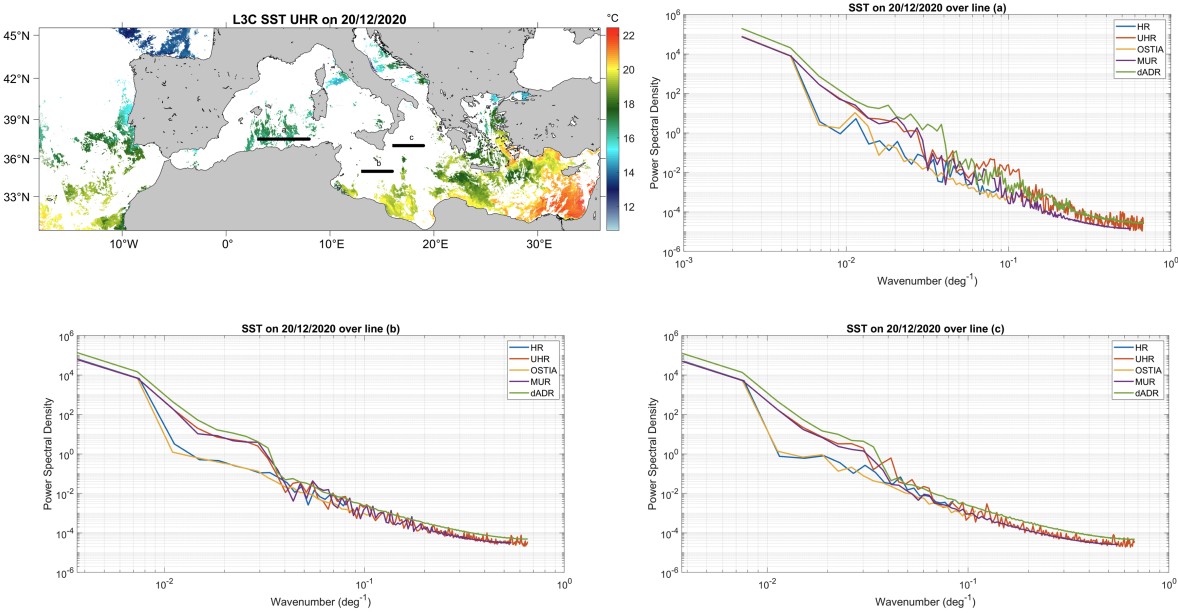

**Figure 9.** The Power Spectral Density profiles (bottom panels) of different SST products under cloudy conditions calculated along the three transects (black lines) on the L3S SST field on 20 December 2020 in the top panel: in blue the MED L4 NRT HR product at 1/16° (Copernicus), in red the MED L4 NRT UHR at 1/100° (Copernicus), in yellow the GLOBAL OCEAN OSTIA product at 0.05° (Copernicus), in purple the Multi-scale Ultra-high Resolution (MUR) product at 0.01° (NASA-JPL) and in green the dADRSR reconstruction developed in this work.

## 3.2 An extensive test dataset

Having established the potential ability of the network, we performed an additional validation test with the aim to strengthen the robustness of the statistics. We built a new totally independent dataset exploiting the L4 SST fields used as initial guess for the OI algorithm which produces gap-free SST maps over the Mediterranean Sea for the year 2021 at $0.01° \times 0.01°$ spatial resolution. Before applying the dADRSR network to this dataset, the fields were preprocessed following the same steps described in Sec. 2.1. The comparison of the evolution of the daily, basin-scale RMSE for the input first guesses and the dADRSR reconstructions for the entire year confirms the improvement of the network output with respect to the first guess fields (Fig. 10). The mean values of the RMSE for 2021 are in line with the ones found for the previous test (i.e., 0.31°C for the dADRSR output versus 0.33°C for the first guess approximation).

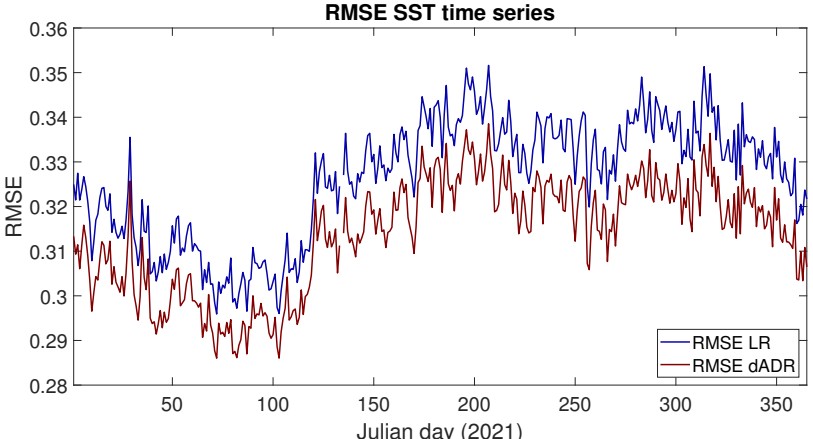

**Figure 10.** Comparison of the daily RMSE time series of the SST obtained via dADRSR reconstruction (red) and the L4 First Guess (blue) with respect to the L3S data measured by Sentinel 3A and 3B satellites during the year 2021. The jump at Julian day 134 is due to the absence of L3 observation for that day which made impossible the comparison with the input and the output of the network.

Analogous to Fig. 6, the differences (averaged on $1° \times 1°$ boxes) between the errors of the two methods approximating the L3S Sentinel observed data (Figure 11), presents an overall improvement of the output of the dADRSR network with respect to the interpolated maps. The red positive values are found almost everywhere for both SST and SST gradients fields.

## 4   Conclusions

The advance obtained by the application of machine learning-based techniques for the improvement of the effective resolution of remote sensing observations have recently opened a new way to approach satellite-derived data processing. The great advantages provided by making high resolution gap-free images available for a wide range of scientific users are severely limited by the number of valid L3 observations. In the case of sea surface temperature measurements, infrared data are commonly contaminated by cloud cover, reducing the quality of the L4 data that can be obtained via statistical interpolation techniques.

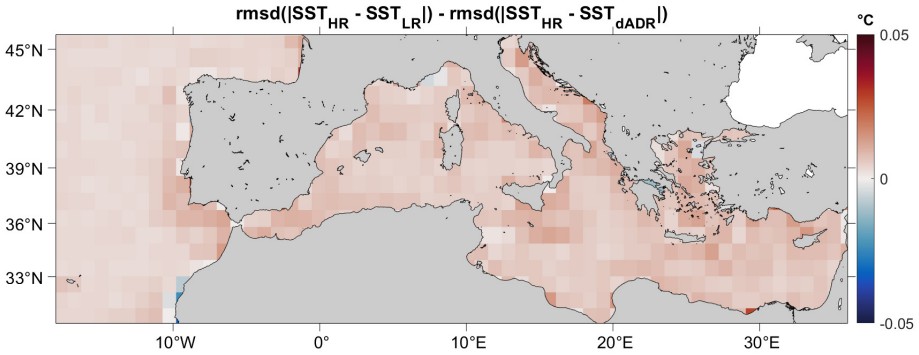

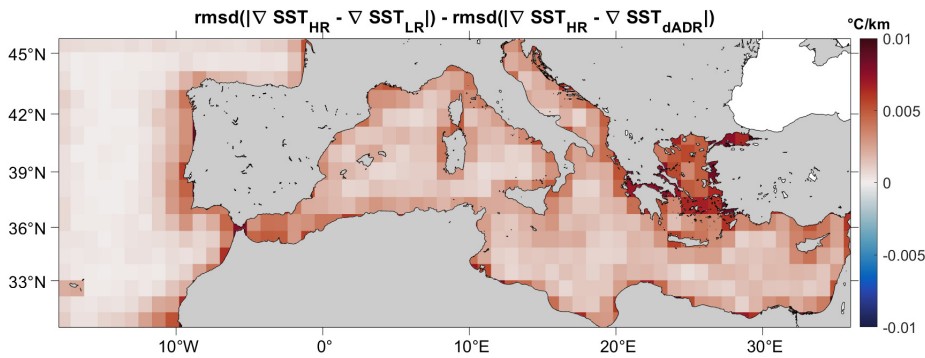

**Figure 11.** Comparison of the performance of the SST (top) and $\nabla SST$ (bottom) dADRSR reconstruction and the L4 First Guess with respect to the L3S data measured by Sentinel 3A and 3B satellites during the year 2021. Red positive values show an improvement of the network reconstruction with respect to the optimal interpolated First Guess.

The machine learning approach used here exploits progress made in the field of computer vision for extrapolating high resolution features even when a direct measurement is missing. Learning directly from L3S SST fields, and taking advantage of both dilated convolution and attention mechanisms, the deep neural network employed here proved able to reproduce small scale signals generally smoothed out by Optimal Interpolation algorithms. The strong variability of the SST in the Mediterranean Sea allowed us to obtain excellent results even considering just one year of data during the training phase. However, it would

be important to investigate whether using longer time series may help to improve the network ability to reconstruct SST fields, as well as to rely on more robust statistics. Moreover, given the inhomogeneity of the spatial error distribution related to the

interpolation technique, it would be interesting to expand the present investigation in order to take into account the OI error field as an additional predictor and to consider the contribution of the error of the SST gradients in the loss function. Another aspect that deserves further investigation concerns the applicability of the dADRSR network to differentiate sea/ocean areas, even though a fine-tuning of the model would probably be needed.

In the future, we also plan to study other super-resolution techniques, which have recently become popular in the field of computer vision. Firstly, we are currently investigating the possibility to improve the reconstruction of small scale features in SST fields via other successful generative AI, such as GANs or diffusion models. The former exploits the outcomes of a minmax game between a generator of reconstructed images and a discriminator which tries to distinguish the real image from the output of the other network; the latter builds super-resolved fields initially introducing noise into the initial signal to then reverses this process until convergences to the desired distribution. However, although such networks applied to remote sensing data have proven to be able to reconstruct very realistic small scale structures, they seem to fail to optimize a point-match evaluation of reconstructed remote-sensed SST fields. Secondly, we would like to explore the usage of Vision Tranformers (ViT) for understanding and reproducing high-level structures by understanding contextual relationships between the patches of an image.

The results achieved here, however, may already benefit a wide range of applications. Super-resolved SST fields would facilitate the challenging task of 2D/3D ocean dynamics reconstruction in synergy with other variables (e.g., Buongiorno Nardelli et al., 2022; Fablet et al., 2023) or the monitoring of ocean fronts in areas of particular interest (e.g., areas affected by vertical exchange and upwelling regions). Enhancement of the effective resolution of SST data and especially SST gradients may also benefit data assimilation in forecast modelling, given their proven sensibility to small structures of sea surface temperature (Maloney and Chelton, 2006; Woollings et al., 2010). We also plan to validate our results exploiting the high resolution SST data derived by the CNN reconstruction within the operational SST chain in the framework of the Copernicus Marine Service.

*Author contributions.* Conceptualization, C.F., A.P. and B.B.N; methodology, C.F., D.C., A.P. and B.B.N.; investigation, C.F., D.C., A.P. and B.B.N.; visualization, C.F.; writing—original draft, C.F.; writing—review and editing, C.F., D.C., A.P. and B.B.N. All authors have read and agreed to the published version of the manuscript.

*Financial support.* This work has been supported by the Copernicus Marine Service, funded through contract agreement no. 21001L03-COP-TAC SST-2300– Lot 3: Provision of Sea Surface Temperature Observation Products (SST-TAC).

*Competing interests.* The authors declare no conflict of interest.

*Acknowledgements.* The authors wish to thank the reviewers for the useful feedback and, in particular, Peter Cornillon for his thoughtful

suggestions, which we think have significantly improved the quality of this work.

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
