# Peer review of "Deep Learning for Super-Resolution of Mediterranean Sea Surface Temperature Fields"

_EGUsphere, 2024_

## Referee Comment (RC1)

The article presents a study on improving the resolution of Sea Surface Temperature (SST) fields in the Mediterranean Sea using deep learning models, specifically a dilated convolutional multi-scale learning network. This approach allows for better capture of small scale features and gradients in SST data, overcoming limitations of traditional satellite-based measurements and interpolation methods. The study demonstrates significant improvements in the accuracy and resolution of SST reconstructions, highlighting the potential of deep learning in enhancing oceanographic data analysis and climate research. But the experiment needs some work.

1. Incorporate additional independent datasets for validating the improved SST fields, ensuring the model's robustness across various conditions and regions within the Mediterranean Sea.

2. Compare the performance of the proposed deep learning model against existing other deep learning super-resolution models, such as GAN series, providing a comprehensive analysis of its advantages and limitations.

3. Conduct a sensitivity analysis to understand the impact of different parameters within the dilated convolutional multi-scale learning network, optimizing the model's performance.

4. Could the article be enriched by including a paragraph discussing how high-resolution SST fields can be incorporated into regional climate models to improve the accuracy of climate projections in the Mediterranean region?

---

## Referee Comment (RC4)

[referee-annotated manuscript omitted]

---

## Author Comment (AC1)

**Manuscript title:** Deep Learning for Super-Resolution of Mediterranean Sea Surface Temperature Fields
**Authored by:** Claudia Fanelli, Daniele Ciani, Andrea Pisano, and Bruno Buongiorno Nardelli
**Manuscript ID:** https://doi.org/10.5194/egusphere-2024-455

**REVIEWER #1**

**Reviewer:** The article presents a study on improving the resolution of Sea Surface Temperature (SST) fields in the Mediterranean Sea using deep learning models, specifically a dilated convolutional multi-scale learning network. This approach allows for better capture of small scale features and gradients in SST data, overcoming limitations of traditional satellite-based measurements and interpolation methods. The study demonstrates significant improvements in the accuracy and resolution of SST reconstructions, highlighting the potential of deep learning in enhancing oceanographic data analysis and climate research. But the experiment needs some work.

**Response:** The authors would like to thank the anonymous reviewer for their interest in reading our manuscript. We think that the revised version of the manuscript has been enhanced significantly as a result of the reviewers' feedback.  Please find the detailed responses below with the reference to the modifications carried out in the re-submitted files (highlighted in yellow).

**Reviewer:** Incorporate additional independent datasets for validating the improved SST fields, ensuring the model's robustness across various conditions and regions within the Mediterranean Sea.

**Response:** We agree with the reviewer that a larger test dataset was needed to provide  a more robust assessment. Therefore, we built a new independent dataset using SST fields over one year (2021) and performed a new test. The main results of this extended test are described in new Section 3.2.

**Reviewer:** Compare the performance of the proposed deep learning model against existing other deep learning super-resolution models, such as GAN series, providing a comprehensive analysis of its advantages and limitations.

**Response:** Following the reviewer suggestion, we have included in the revised text a comparison between the results obtained with the dADR model and other deep baseline computer vision methods.  The outcomes are now presented in Table 1. We are currently working on the application of Generative Adversarial Networks to the problem, but our study is still in a preliminary phase and does still require a substantial effort to be ready for publication. As such, considering also some subtle theoretical aspects related to the application of generative approaches to observational data, we prefer to dedicate an entire new scientific article to discuss/present our findings, as now discussed in lines 321-328 of the revised version of the manuscript.

**Reviewer:** Conduct a sensitivity analysis to understand the impact of different parameters within the dilated convolutional multi-scale learning network, optimizing the model's performance.

**Response:** Thanks to the reviewer's comment, we realized that we did not explain why we had chosen a specific configuration of the network, and we also understood that at least one additional test was needed. In fact, we did not develop a new architecture but exploited the one developed by Buongiorno Nardelli et al. (2022) (https://doi.org/10.3390/rs14051159), and also aimed to reduce computational costs. As such, we originally relied on the choices made in that specific work . In the revised paper, however, we now describe one additional test carried out by modifying the network depth, i.e. reducing the number Multiscale Adaptive Residual Blocks (called dADRSR/2 in the revised version of the manuscript) and compare that also with the results obtained by other deep learning methods (i.e., the EDSR and the ADR networks). The assessment of the various network configurations considered are now presented in Table 1.

**Reviewer: Could the article be enriched by including a paragraph discussing how high-resolution SST fields can be incorporated into regional climate models to improve the accuracy of climate projections in the Mediterranean region?**

**Response:** To our knowledge, using observation-based SST data is not a standard procedure to improve the accuracy of climate projections, which generally do not rely on assimilation strategies. High resolution SST fields are indeed used for the improvement of operational ocean model re-analyses (e.g. within Copernicus Marine Service), which tackle a very different problem with respect to climate projections. Even if there is evidence of the important role of small structures of SST on different processes that are clearly relevant in modulating climate (e.g. atmosphere-ocean interactions, see Renault, et al. "Modulation of the Oceanic Mesoscale Activity by the Mesoscale Thermal Feedback to the Atmosphere". *J. Phys. Oceanogr.* **53**, 1651–1667, 2023), we feel that introducing a specific discussion on this topic would be distracting and not relevant for the readers.

---

## Author Comment (AC2)

**Manuscript title:** Deep Learning for Super-Resolution of Mediterranean Sea Surface Temperature Fields
**Authored by:** Claudia Fanelli, Daniele Ciani, Andrea Pisano, and Bruno Buongiorno Nardelli
**Manuscript ID:** https://doi.org/10.5194/egusphere-2024-455

**REVIEWER #2**

**Reviewer:** **This study aims at reconstructing small scales SST from a low resolution SST field and provides a gap free L4 dataset resolving scales up to 5 km using deep learning algorithms. The super Resolution Convolution Network is learning using an ensemble of low resolution and high resolution SST images in the Mediterranean Sea. Results are shown for one snapshot and analyses are shown for cloud-free areas. Compared to the first guess at low resolution, the SST field has indeed been improved but this study needs further investigation and discussion of the results.**

**Response:** The authors would like to thank the anonymous reviewer for their interest in reading our manuscript. We feel that the revised version of the manuscript is significantly enhanced as a result of the reviewer's feedback. Please find the detailed responses below with the reference to the modifications carried out in the re-submitted files (highlighted in yellow).

**Reviewer:** **Comparison with other High resolution L4 products are required as the first guess is at very low resolution and does not reflect what is currently available (such as MUR product from JPL for example). Some SST products perform well when the cloud coverage is small but have a resolved spatial scale that varies a lot with the cloud cover. Comparison of the different SST products on a cloud-free event and on a cloudy day will show the potential of this new SST product much more homogeneous in time.**

**Response:** First of all, thanks to the reviewer's comment, we realized that our objective and the super-resolution workflow within the CNR processing chain could be better introduced. In fact, CNR 1 km grid processing chain is based on a two step algorithm. The first step provides a low resolution field that is needed to obtain robust SST estimates below large cloudy areas, that is successively used as the background field for the application of a more local optimal interpolation algorithm. This background/first guess field is generally quite smooth even in cloud-free areas, and our objective is to improve its effective resolution and the intensity of the related gradients (it is thus an intermediate step in the processing chain). As such, the output of the network is not a new product itself, as clarified in Figure 1 of the revised version of the manuscript.
Then, we agree with the reviewer that it is quite useful and relevant to intercompare our original first-guess and the super-resolved one also with other products. Consequently, we performed a power spectral density analysis on different SST products available over the Mediterranean Sea (see Figure 9 and lines 278-291 of the revised version of the manuscript). The products used are the L4 NRT HR and the L4 NRT UHR provided by CNR for the Copernicus Marine Service, the GLOBAL OCEAN OSTIA product developed by the UK MET OFFICE, the Multi-scale Ultra-high Resolution (MUR) product provided by the NASA-JPL and the super-resolved SST field obtained by the application of the dADR-SR network developed in this work. We focus on cloudy areas and not on cloud-free events as this is where super-resolution is expected to provide an improvement.

**Reviewer:** **From the snapshot with the SST gradient, One wonders if the method enables the reconstruction of smaller scales or is more of a gradient enhancement. It would be helpful to comment on that and illustrate if submesoscales are really generated in this SST reconstruction.**

**Response:** The reviewer is absolutely right: this specific algorithm is mostly enhancing the gradients and it is in fact more conservative in the reconstruction of small scale features. Indeed, other methodologies (those based on Generative Adversarial Networks) have the potential to re-create small scale eddies/filaments which would be compatible with the learned features. However, single-image end-to-end approaches might not be the best choice to obtain accurate and dynamically consistent reconstructions (though surely looking quite "realistic"). Indeed, we are presently carrying out tests with GAN, but considering the time-frame and the amount of work still needed to get to (at least partly) conclusive results, those will necessarily need to be described in a successive paper.

**Reviewer:** **The added value of this SST product is not clear from Table 1 and the spectrum figure (Fig.6). I would recommend adding other SST high resolution L4 products as discussed in point 4. Regarding the SSIM index, it may be interesting to also discuss the details of the decomposition between contrast / structures and luminance if the results prove to be relevant (the contrast and structure are expected to be much more improved than the luminance).**

**Response:** As also explained above, to demonstrate how the super-resolved images reach an enhanced spatial resolution reconstructing SST fields under cloudy conditions, we performed a power spectral density analysis on different SST products available over the Mediterranean Sea. This is now fully documented in Figure 9 and lines 278-291 of the revised version of the manuscript. Given the fact that the SSIM index is not a common metric used in oceanography but it is mainly used in the computer vision field, we feel that is out of the goal of this paper to discuss the details of the single parts of this measure.

**Reviewer:** **Finally, to really describe the effective resolution, the study of the ratio between the spectral content of the reconstructed data and the truth is more relevant than mere spectrum (as detailed in Ballarotta, et al 2019.). It should be included in a further analyses of this SST product to comment on the effective resolution of the SST product.**

**NB: Ballarota et al 2019: Ballarotta, M., Ubelmann, C., Pujol, M.-I., Taburet, G., Fournier, F., Legeais, J.-F., Faugère, Y., Delepoulle, A., Chelton, D., Dibarboure, G., and Picot, N.: On the resolutions of ocean altimetry maps, Ocean Sci., 15, 1091–1109, https://doi.org/10.5194/os-15-1091-2019, 2019.**

**Response:** The reviewer is absolutely right.. We now added the computation and the analysis of the PSD ratio as defined in Ballarotta et al. (2019) in lines 273-277 and Figure 8 of the revised version of the manuscript.

---

## Author Comment (AC3)

**Manuscript title:** Deep Learning for Super-Resolution of Mediterranean Sea Surface Temperature Fields
**Authored by:** Claudia Fanelli, Daniele Ciani, Andrea Pisano, and Bruno Buongiorno Nardelli
**Manuscript ID:** https://doi.org/10.5194/egusphere-2024-455

**REVIEWER #3**

**Reviewer:** In their work the authors are tackling an important problem in geophysics, namely the reconstruction of high resolution fields of Sea Surface Temperature from partial high-resolution observations. Super-resolution is an ill posed problem, given that identical low-resolution fields can correspond to different high-resolution fields. The authors adapt their previous work on ADT to SST field reconstructions, as well as their well established knowledge of SST field reconstruction, showing improvements over the Mediterranean basin, within the confines of their experiment.

I really appreciated many parts of the article, notably the varying metrics and case studies to evaluate the quality of the reconstructions.

As it stands, I have some major and minor criticisms for the article that, should the authors address, would make for a significant contribution to the community.

**Response:** The authors would like to thank the anonymous reviewer for their interest in reading our manuscript. We think that the manuscript has been significantly improved by taking into account the reviewer's feedback. Please find the detailed responses below with the reference to the revisions appearing in the re-submitted files (highlighted in yellow).

**Reviewer:** The validation process is prone to data leakage. 4 days out of a year of data were omitted, but there is no mention of removing some days before or some days after in order to prevent data leakage. The physical reasoning of this is absent. Are the structures that decorelated after one day given the removal of the 200km smoothed field?

In general the L119 statement: "The test dataset is finally selected separating the 15% of the tiles available after 120 the preprocessing, chosen in order to be able to reconstruct the full geographical coverage of four days which are representative of different seasons." requires clarification. I read it as 4 individual days, one in each season. It could be understood as patches covering the whole area, spread out over each season. I would expect to have more of a cross-validation approach given the one year dataset limitation.

**Response:** Thank you for pointing out the lack of clarity and previous limitations regarding the choices made to construct the test dataset. Indeed, the initial test was carried out selecting all the tiles covering four individual days (one in each season), as clarified in line 132 of the revised version of the manuscript. However, we do agree about the limitations of that initial test dataset and thus set up a much more extended test including one full year of totally independent data. The new results are presented in the new Section 3.2.

**Reviewer:** **Another major concern is the input. The input, presently, is only the first guess (removing a sliding window), and the information coming from the L3 satellite product is not used as a complimentary input. Why did the authors deprived themselves from potential additional input such as multiple time steps and L3 products? Other works (such as Archambault et al, Martin et al) in SSH fields have training procedures where some of the satellite information is omitted from the target in order to validate the approach.**

**Response:** Thanks to the reviewer's r comment, we realized that we needed to better introduce the objectives and processing steps involved in our workflow . In fact, we are not dealing here with an algorithm designed for gappy-fields interpolation, but rather with an algorithm designed to improve one specific step in our processing chain, namely improving the resolution of the low-resolution gap-free field used as the background for the 1 km optimal interpolation implemented in our chain. As such, unlike the classical interpolation methods, in which it is important and useful to use ground-truth data and multiple time steps as input when available, the goal here is effectively only to learn how to reconstruct a single image at very high resolution starting from a low resolution one. L3 data are indeed e used as input in the final interpolation step, though, and are of course also used as target during the training phase. We have clarified all this in the Introduction of the revised version of the manuscript (lines 87-91).

**Reviewer:** **There is no mention of how the total field reconstruction over the whole Mediterranean sea is output. If the image was made by recomposing a sliding window reconstruction that should be mentioned. Given that the network learns filters, one could conceivably apply them on the whole image, though I expect the attention layers to pose an issue.**

**Response:** Following from previous comment/response, we hope to have clarified this point adding a more detailed explanation at the end of Section 2, in lines 208-212 of the revised version of the manuscript.

**Reviewer:** **The choice of architecture, while documented, is not justified. Were the hyper-parameters optimized? Other architectures evaluated? The authors mention a lot of competing methods, but do not compare their architecture to them. (DINEOF, DINCAE, to cite but two) Are the computational and expertise cost justified versus other methods? The results seem to indicate a 0.02°C improvement; is the architecture stable through different initializations?**

**Response:** As clarified in the revised text, we are not dealing here with a novel interpolation algorithm, but rather on a single-image end-to-end algorithm, that is applied to improve our background field used in a two-step optimal interpolation algorithm. As such, comparisons with different interpolation algorithms as DINCAE/DINEOF would be out of scope. On the other hand, we realized that we did not explain why we have chosen a specific configuration of the network, and we also understood that at least one additional test was needed. In fact, we did not develop a new architecture but we exploited the one developed by Buongiorno Nardelli et al. (2022) (https://doi.org/10.3390/rs14051159), and also aimed to reduce computational costs. As such, we originally relied on the choices made in that specific work. In the revised paper, however, we now describe one additional test carried out by modifying the network depth, i.e. reducing the number of Multiscale Adaptive Residual Blocks (called dADRSR/2 in the revised version of the manuscript) and

compare that also with the results obtained by applying other deep learning methods (i.e., the EDSR and the ADR networks) The assessment of the various network configurations considered are now shown in Table 1.

**Reviewer:** **The method section (2.2) seems to assume unfamiliarity with neural networks, providing intuitive explanations for basic architectural blocks, but then very quickly skims over important details of the more complicated blocks of the architecture. This part would benefit reducing the initial explanation of activation functions and CNNs (such as the interpretation of lines 59 to 61 which is intuitive but could easily not correspond to the exact explanations provided given the non-linear activations) and expanding on the reasoning of the architectural choices (the adaptive part of the ARB is not discussed, implying the rest of the architecture is non adaptive).**

**Response:** This has now been revised considering the reviewer suggestion. We clarified the adaptive strategy used in the network in lines 72-75 and lines 161-164 of the revised version of the manuscript.

**Reviewer:** **No mention is made to VIT and diffusion-based super resolution techniques that have become state of the art in computer vision. I can understand the daunting nature of these, but should they not be mentioned as potential further steps, at least? The latter is especially significant: the field reconstructions obtained through optimizing RMSE favor smoothness, and often do not represent physically feasible oceanic states. Graphcast for example has been abandoned in favor of Gencast for that very reason. Given that the model is in NRT, and therefore would be used for constraining operational models, it might be interesting to at least think about this. It is even more important given the non-bijective nature of the problem.**

**Response:** We agree with the reviewer that it is worth mentioning also other techniques that have gained a lot of interest lately, which we would like to explore in the future. Following their suggestion, we added a brief paragraph in the conclusion on the topic (lines 321-320 of the revised version of the manuscript).

**Reviewer:** **Fig.1 would benefit from locating with a bounding box or three the patches on the right hand side.**

**Response:** Thank you for your suggestion. We modified Figure 2 of the revised version of the manuscript with an example of a pair of tiles and the corresponding position over the Mediterranean Sea.

**Reviewer:** **182: max(I) in denormalized space i.e. K°? Or in the space where the large field is removed 200km? Is it computed over the patches, or the whole Mediterranean?**

**Response:** All the errors (including the max(I) used to calculate the PSNR) are computed over the final reconstructed image over the whole Mediterranean Sea, as explained in lines 208-212 of the revised version of the manuscript.

---

## Author Comment (AC4)

**Manuscript title:** Deep Learning for Super-Resolution of Mediterranean Sea Surface Temperature Fields
**Authored by:** Claudia Fanelli, Daniele Ciani, Andrea Pisano, and Bruno Buongiorno Nardelli
**Manuscript ID:** https://doi.org/10.5194/egusphere-2024-455

**REVIEWER #4 (Peter Cornillon)**

**Reviewer:** This manuscript explores the use of a deep learning model to enhance the spatial resolution of L4 SST products in regions where missing data, generally associated with cloud cover, results in coarse fields obtained with objective analysis techniques. As I understand it, the model, which the authors have developed, dADR-SR, is trained with high resolution (HR, 1/16 degree) input fields and ultra high resolution (UHR 1/100 degree) target fields. It is then applied to HR fields and shown to reproduce structure at very nearly the same spatial resolution as test (1/100 degree) fields.

**Response:** The authors would like to thank Peter Cornillon for his interest in reading our manuscript and for his thoughtful suggestions. We think  that the manuscript has been significantly improved by taking into account his feedback .  Please find the detailed responses below with the reference to the revisions made in the re-submitted files (highlighted in yellow).

**Reviewer:** The ML model they have developed appears to perform very well for the test dataset they use but I struggled with the manuscript and I believe that it needs a fair amount of editorial work before it is ready for publication. Specifically, after reading the manuscript several times, I think that I sorted out what was done but, I must admit that I am still not sure that I have it right. I've included a figure in which I have tried to show the datasets, which I think they are using, and how these dataset relate to one another. First, there is a set of four datasets, two high resolution (1/16 degree), an L3S and an L4, and two ultra high resolution (1/100 degree) again an L3S and an L4. A subset of these datasets are used to train the dADR-SR algorithm and the HR L4 dataset is then fed into the trained model and the output is compared with an L3S UHR dataset (i.e., one built using the same algorithm as used to build the standard L3S UHR products used to train) constructed with SLSTR data from the Sentinel 3A and 3B satellites. I don't think that the SLSTR data are used in the construction of the standard products but I may have that wrong, well, I may have all of this wrong, for which I apologize. Adding to the confusion is that the authors appear to have changed the terminology they use for the datasets. In the abstract and in the Discussion section the authors refer to low resolution (LR) and high resolution (HR) datasets while in the remainder of the document they refer to high resolution (HR) and ultra high resolution (UHR) datasets. I'm guessing that LR (used in the Introduction and Discussion)  is what they later refer to as HR and HR (in the Intro and Discussion) is what they later refer to as UHR.

**Response:** Thank you for pointing out the lack of clarity of the characterization of the different datasets used and also of the related processing chain steps. To address this point, we have added a more comprehensive description of the methodology used for each

dataset, at in lines 101-114 of the revised version of the manuscript, added the new Figure 1 and removed the acronym "LR" when referring to the first guess background field, due to the clear confusion it was creating throughout the manuscript.

**Reviewer:** **Bottom line: I believe that a bit more description of what goes into the standard datasets that are used to train and later as input to the dADR-SR model, along with a clear description of differences, if any, between the datasets produced as input and/or evaluation for the work undertaken in this study. I also think that a figure showing the relationship of the datasets and processing steps, at a very gross level—sort of like the figure that I have attempted to put together below—would go a long way to making the manuscript easier to follow.**

**Response:** Thank you for your suggestion.We do agree that a much more clear description of the methodology behind the creation of the datasets was necessary. We provided all these details at lines 101-114 and put a sketch describing the workflow as the new Figure 1.

**Reviewer:** **In addition to the general concern outlined above I have made a number of editorial suggestions, which I hope will help to make the manuscript a bit easier to read. These are included in the attached manuscript either as hand-written annotations or as typed comments. Finally, I would like to apologize to the authors for the length of time that I took for this review—I had another manuscript, which I was asked to review, and which had to be completed before addressing this one as well as some family issues.**

**Response:** We really appreciate and do  thank Peter Cornilon for the attention paid to revise this manuscript. We made all corrections following  his suggestions, as highlighted in yellow throughout the revised version of the manuscript.

---

## Referee Report (RR1)

[referee-annotated manuscript omitted]

---

## Author Response (AR2)

**Manuscript title:** Deep Learning for Super-Resolution of Mediterranean Sea Surface Temperature Fields
**Authored by:** Claudia Fanelli, Daniele Ciani, Andrea Pisano, and Bruno Buongiorno Nardelli
**Manuscript ID:** https://doi.org/10.5194/egusphere-2024-455

**REVIEWER #4 (Peter Cornillon)**

**Reviewer:** First, my apologies to the authors for the delay in providing this review. Given that it is the second review, I thought that I could complete the review relatively quickly. However, I struggled with this version and I think that it still needs some editorial work. In my first review, the most significant concern I raised related to confusion that I had with regard to the products the authors were using. The authors have made a significant attempt to address this by more carefully describing the basic datasets. Thanks for this, the figure that you have added was particularly helpful. This addressed my concern with regard to the input datasets but I still found the text to be confusing. I think that the issue is that a number of datasets are being used and these are not referenced consistently in the manuscript. My sense is that a rather straightforward solution to this problem would be to adopt a consistent naming convention across all of the datasets and then to use this throughout. Specifically, I suggest something like: L3S HR, L4 HR, L3S UHR, L4 UHR and L3S Sentinel. There actually may be a couple of other data sets, like the First Guess one, which is different from these.

**Response:** We have followed your suggestions in the revised version of the manuscript and we now reference the datasets as L3S HR, L4 HR, L3S UHR, L4 UHR, L3S Sentinel and First Guess. The network is called dADRSR.

**Reviewer:** I think that I'm still confused about how the First Guess (is this the same dataset as the Super-resolved First Guess shown in Fig. 1?), Low Resolution and OI datasets differ, if they do. The authors seem to use these terms interchangeably. If they are the same, then choose one name and use it throughout. Furthermore, I would choose a name that works with the ones that I suggest above if at all possible.

**Response:** Indeed we did use the terms "First Guess", "Optimal Interpolated Field" and "Low Resolution Field" for the same product. Given the confusion that we created by doing it, we now refer to this dataset as "First Guess" maps.

**Reviewer:** The use of L3 is not consistent throughout either—in at least one case a dataset is referred to as L3, L3S and L3C, at least I think that it is the same dataset. This just adds confusion to the text.

**Response:** In the revised version of the manuscript, we now refer to L3S HR or L3S UHR if recalling the two datasets produced for the Copernicus Marine Service and to L3S Sentinel when describing the target dataset used for the training and the validation of the convolutional neural network.

**Reviewer:** In addition to the general concern outlined above I have made a number of editorial suggestions, which I hope will help to make the manuscript a bit easier to read. These are included in the attached manuscript either as hand-written

**annotations or as typed comments.**

**Response:** Thank you for all your suggestions, we followed them and modified the manuscript according to your feedback.